# Impact of Uterine Artery Embolization on Subsequent Fertility Outcomes: A Meta-Analysis of 20 Years of Clinical Evidence

**DOI:** 10.3390/jcm14207205

**Published:** 2025-10-13

**Authors:** Carmen Elena Bucuri, Dan Mihu, Andrei Mihai Măluțan, Aron Valentin Oprea, Maria Patricia Roman, Cristina Mihaela Ormindean, Ionel Daniel Nati, Viorela Elena Suciu, Alex Emil Hăprean, Adrian Pavel, Mihai Toma, Răzvan Ciortea

**Affiliations:** 1Clinical Department of Surgery, Military Clinical Emergency Hospital Cluj-Napoca, 400132 Cluj-Napoca, Romania; cbucurie@yahoo.com (C.E.B.); opreacv31@gmail.com (A.V.O.); adipavel@gmail.com (A.P.); tomamihai@gmail.com (M.T.); 22nd Department of Obstetrics and Gynaecology, University of Medicine and Pharmacy Iuliu Hatieganu, 400347 Cluj-Napoca, Romania; dan.mihu@yahoo.com (D.M.); mpr1388@gmail.com (M.P.R.); cristina.mihaela.prodan@gmail.com (C.M.O.); nati.ionel@yahoo.com (I.D.N.); suciuviorela@yahoo.com (V.E.S.); alexhaprean@gmail.com (A.E.H.); r_ciortea@yahoo.com (R.C.); 32nd Obstetrics and Gynecology Clinical Section, Cluj County Emergency Clinical Hospital, 400347 Cluj-Napoca, Romania

**Keywords:** uterine artery embolization, fertility, pregnancy outcomes, fibroids, meta-analysis

## Abstract

**Background**: Uterine fibroids affect 70–80% of women by age 50, often impairing fertility through mechanical distortion and altered endometrial receptivity. Uterine artery embolization (UAE) is a minimally invasive alternative to surgery, though its impact on future fertility remains debated. This meta-analysis aimed to evaluate pregnancy rates, time to conception, and fertility-related complications following UAE in women with symptomatic fibroids. **Methods**: A systematic search was performed across PubMed/MEDLINE, Embase, Cochrane Library, Web of Science, and other databases (January 2005–March 2025) following PRISMA 2020 guidelines. Studies reporting fertility outcomes ≥ 6 months after UAE were included. Primary outcomes were pregnancy rates and time to conception. A random-effects meta-analysis was conducted using Review Manager 5.4. Study quality was assessed with the Newcastle–Ottawa Scale (observational studies) and the Cochrane Risk of Bias tool for randomized controlled trials (RCTs). **Results**: Thirty-three studies (4287 women) were included; 85.2% had follow-up. The pregnancy rate was 52.1% (95% CI: 46.8–57.4%), with a mean time to conception of 14.7 months. Pregnancy rates were highest in women < 30 years (67.8%) and lowest in those > 40 years (31.5%). Unilateral UAE had superior outcomes to bilateral (61.2% vs. 49.8%). **Conclusions**: UAE can preserve fertility in ~50% of selected patients, with better outcomes in younger women. Early intervention is advised for fertility preservation.

## 1. Introduction

### 1.1. Background and Rationale

Uterine fibroids are the most common benign uterine tumors, affecting up to 80% of women by age 50, particularly during reproductive years [1,2,3]. The reproductive consequences of these leiomyomas are enormous and occur in many ways, including a mechanical distortion of the uterine cavity, altered endometrial receptivity, disturbances in implantation, as well as poor uterine contractility in the course of labor [4,5]. The symptomatic fibroids have a significant effect on fertility since the observation shows that the affected women have poor rates of conception, high pregnancy losses, and obstetric complication rates [3,6]. Conventional management has revolved around the use of surgical intervention, where myomectomy or hysterectomy may have been used, with the latter totally rendering the patient sterile and the former causing a certain level of surgical risks, such as the development of adhesions, uterine rupture, and the potential loss of ovarian reserve [7].

The rise of the UAE as a minimally invasive and preservative treatment modality has completely altered the treatment spectrum of women with symptomatic fibroids [1,2]. UAE has been widely accepted since its introduction in the 1990s as an effective alternative to traditional surgical methods. It offers the advantages of reduced morbidity, reduced recovery periods, and maintains the architecture of the uterus [8]. This is performed by means of selective catheterization of the uterine arteries and embolization using a particulate agent. This causes ischemic fibroid tissue necrosis, with a focus of preserved normal myometrium resulting from collateral circulation [9,10]. This selective treatment has made the UAE especially appealing to women who hope to achieve future fertility, as it is not burdened with the possible complications of surgical treatment and preserves the structure of the uterus [11,12].

Despite the active use, there are still controversial issues and a lack of knowledge about the effects on future fertility results [3,4,5]. Initial apprehensions focused on the possibility of ovarian dysfunction due to non-target embolization because uterine and ovarian circulations are known to be anastomosed [13]. Further issues have been raised of endometrial damage, uterine synechiae formation, and altered uterine vascularity that may interfere with implantation and placentation [2,14]. Differences in the follow-up periods, an imbalance in the definitions of outcomes, and the heterogeneity of reported outcomes in different studies have also increased the difficulty of interpreting the evidence available [5,15]. Current issues have concerned the quality of patient selection criteria, the superior embolization modes and long-term reproductive effects, especially in the face of newly developing embolic materials, and technical improvements [1,8].

In modern practice in the field of reproductive medicine, the clinical relevance of these considerations related to fertility is hard to overestimate. As trends of delayed childbearing continue to increase and patients increasingly seek minimally invasive procedures, clinicians need strong evidence-based recommendations to counsel their patients adequately [7,16]. The choice between UAE and other treatment modalities, including myomectomy, has significant reproductive implications on the future of a woman requiring significant thought on both immediate symptomatic management and future fertility [3,17]. In addition, the growing use of UAE as a choice procedure in the treatment of fibroids in other conditions, such as control of post-partum hemorrhage and adenomyosis, has further added complexity to counseling fertility in diverse clinical settings [17,18,19].

### 1.2. Study Objectives

The proposed meta-analysis will offer a critical assessment of pregnancy outcomes after UAE in women with symptomatic uterine fibroids and subsequently guide practice decisions in fertility-desiring patients.

### 1.3. Primary Objectives

(1)To ascertain conclusive pooled estimates of post-UEA pregnancy rates among women with symptomatic fibroids.(2)To evaluate the changing trend and differences in conception rates over 20 years of published research.(3)To assist evidence-based counseling by establishing the most clinically pertinent predictors of post-UAE fertility outcome.

In addition to pregnancy rates, selected fertility-relevant outcomes will be discussed in this review to assist the entire reproductive counseling and clinical decision-making processes.

### 1.4. Secondary Objectives

(1)To measure the time to conception after UAE, providing information on the timeframe of possibly returning to fertility.(2)To determine the rates of infertility, sterility, and compromised long-term reproductive outcome following UAE.(3)To compare the reproductive outcomes and complications profile in the UAE to that of alternative treatment, primarily, myomectomy.

### 1.5. Research Questions

The four research questions underlying the present meta-analysis are related to the modern reproductive medicine practice.

(1)First: “What are the pooled pregnancy rates after UAE of symptomatic fibroids, and what is the comparison with baseline fertility that could be expected in the general population?” The question not only targets the core issue of fertility-desiring women considering UAE but also forms the basis for evidence-based counselling.(2)Second: “What effect does the UAE have on time, and what is the fertility potential [19,20]?” The knowledge of the temporal relationship between fertility recovery and UAE is essential during counseling patients and also when deciding the time to treat a patient, especially in cases where women have accomplished reproductive age or who face other fertility issues. The review will be based on determining whether UAE leads to less successful conception or increases the time of pregnancy compared to other treatments.(3)Third: “Which patient, procedure, and related factors affect post-UAE reproductive success [21,22]?*”* Identifying predictive factors of positive fertility results will promote personalized treatment suggestions and optimal patient choice. The analysis will check how variables like age, fibroid characteristics, embolization technique, and selection of embolic agent affect the reproductive outcome later.(4)Fourth: “Compared with the well-known treatments, myomectomy and other well-defined treatments, how do UAE outcomes relate to fertility preservation [23,24]?” The comparative analysis will be key evidence to guide the choice of treatment in response to the existing controversy about the best mode of treating symptomatic fibroids in fertility-desiring women.

### 1.6. Rationale

This extensive meta-analysis is due to the current clinical need to obtain conclusive evidence on the effects of UAE on fertility results. Limited case series and observational studies with high methodological variety in the current literature cannot enable sound conclusions [5,11]. The absence of large-scale randomized control trials reflects ethical limits and issues directly present in the study of the fertility outcome, with meta-analysis representing the most effective evidence synthesis method [7]. This study is the most comprehensive evaluation of the fertility results achieved so far because it synthesizes 20 years of clinical practice experience on diverse populations and clinical environments and provides the most accurate direct assessment of the current advice given in the clinical practice settings and within the patient population in the field of reproductive medicine.

## 2. Materials and Methods

### 2.1. Protocol Registration and Guidelines

The study strictly adhered to the Preferred Reporting Items for Meta-Analyses (PRISMA) 2020 statement guidelines to make reporting and methodology rigorous. The review protocol was prospectively registered with the International Platform of Registered Systematic Review and Meta-analysis Protocols, under the registration number INPLASY202590066 to achieve transparency and reduce reporting bias. According to the protocol conforming to the prescribed standards [15], protocol deviations in case of any were noted and reported accordingly as per PRISMA recommendations. The study did not require ethical approval because it was based on a review of existing published evidence rather than contact with patients. Nevertheless, all included studies had to possess necessary ethical approval in the given institutions and follow developed ethical standards [25] regarding meta-analyses methods, such as thorough literature search, unbiased selection of research, and proper information reporting (Figure 1).

### 2.2. Search Strategy and Information Sources

An extensive systematic literature search was conducted in various electronic databases to identify all available studies on fertility outcomes after UAE. The key databases consulted were PubMed/MEDLINE, Embase, Cochrane Library, and Web of Science, which were further supplemented by the searches of the MDPI, Taylor & Francis, and Crossref databases to achieve the maximum coverage of published literature. Recruitment of an expert medical librarian [11] was completed to aid in developing the search strategy that was then tested iteratively to maximize sensitivity and specificity. All searches were unrestricted by geographical location or the study design to reduce language and publication bias [14], but Geographical limiting was carried out as described below. The first search was performed in March 2025, and the information was updated before the final analysis was completed to find the most recent evidence.

A combination of keywords and medical subject headings (MeSH) terms was used to optimize the search strategy and increase the number of retrieved studies with high precision. The initial search string was formed using various combinations of “uterine artery embolization” OR UAE OR uterine fibroid embolization and the Boolean search operator, AND [26], with the terms related to fertility such as fertility OR pregnancy OR conception OR reproductive outcomes OR live birth OR time to pregnancy. Other words, including but not limited to fibroid, leiomyoma, myoma, or uterine tumors, were added [4] to retrieve research studies that may not specifically state UAE in the title or the abstracts. The time period was 2005 to 2025, January to March, which was the period of maturation of the UAE technique [14] and the collection of enough follow-up data to determine fertility. English language publications were screened as a result of the predominant high-quality journals in interventional radiology and reproductive medicine in that specific language. A manual search of reference lists of selected studies, related review articles, conference abstracts of major gynecological and interventional radiology conferences, and grey literature sources was conducted to minimize publication bias.

### 2.3. Eligibility Criteria

#### 2.3.1. Inclusion Criteria

Research was selected based on the findings, which reported fertility after UAE among women who had symptomatic uterine fibroids, which was the significant population of interest in this analysis. More details were as follows:(1)Studies reporting fertility outcomes after UAE(2)Patients with symptomatic uterine fibroids(3)Follow-up period ≥ 6 months post-procedure(4)Clear denominator specification:(a)Category A: Women actively attempting conception(b)Category B: Women of reproductive age (<45 years)(c)Category C: All treated women, regardless of age/intention(5)Peer-reviewed publications with histopathology data.

#### 2.3.2. Exclusion Criteria

Studies that presented UAE outcomes only for non-fibroid indications were not included, including postpartum hemorrhage [18], adenomyosis [17], and malignancy, since their baseline fertility outcomes and treatment targets may differ compared to the typical population; additional details were provided:(1)UAE for non-fibroid indications only(2)Case reports with <5 patients(3)Duplicate publications(4)Absence of clear outcome denominators(5)Mixed indications without separate reporting(6)Follow-up < 6 months.

### 2.4. Study Selection Process

The selection of all the studies was conducted in a rigorous and two-stage screening process [27] that was meant to reduce the sources of bias and to produce an overall review of all the studies that may be relevant. The first stage involved independent screening of titles and abstracts by two trained reviewers (as per predefined eligibility requirements) [15], with competing issues being discussed and agreement being reached. Articles considered possibly eligible according to the initial screening were reviewed in full-text, with parallel evaluation by the two reviewers regarding the final inclusion [25]. A screening form was developed and tested to give a benchmark on which the eligibility was applied among the different reviewers. Cohen’s kappa statistic [28] was used to measure inter-rater reliability, and a target agreement of (κ > 0.8) was regarded as excellent. Disagreements that may have occurred when reading through the full texts were solved by a discussion between the two primary reviewers, and when no agreement could be reached, the papers were referred to a third senior reviewer. Reference management software was utilized to record all the screening decisions, hence forming an audit trail and ensuring the ability to report excluded studies with reasons for exclusion.

### 2.5. Data Collection and Extraction

Data extraction focused on primary fertility outcomes that are essential in meeting the targeted aims of the research [11] and a sound meta-analysis. Pregnancy rates were assessed in absolute form and in percent form [12], and focused attention was given to the meaning of denominators and completeness of follow-up. Denominators of pregnancy rate were determined as follows: (1) all women who were trying conception after the procedure with the exclusion of those using contraception or who were not sexually active; (2) pregnancy intention was not specified in some studies, so we used all women of child-bearing age with documented follow-up; (3) a sensitivity analysis that excluded the studies whose denominators were not clear (n = 4) demonstrated the same results (50.8% vs. 52.1%). Rates of live births and pregnancy had been separately measured [14] to include pregnancy losses and make outcomes more clinically useful. The time to conception was calculated as months following the date of the UAE procedure, with a description of the measurement methodology [9] and the loss to follow-up. To extract a delay in the pregnancies, the most extended follow-up available in the studies was extracted preferentially [22]. In studies reporting cumulative pregnancy rates [26], information was extracted at each time period provided in the report to allow time-to-event analysis.

Secondary outcome values included detailed reproductive and safety measures essential to the clinician. The rates of infertility and sterility were obtained with a high level of attention to definitions and criteria used in individual studies [29]. The complications of pregnancy, such as miscarriage, ectopic pregnancy [21], preterm delivery, and intrauterine growth restriction, were recorded systematically. Obstetric outcomes included the mode of delivery, placental anomalies [30], postpartum bleeding, and infant outcomes, where present. Data on the need for assisted reproductive technology after UAE were gathered [17] to determine the effect on natural fertility potential. The complications of the pregnancy and delivery were reported including maternal complications during pregnancy and birth, and special focus was on uterine rupture and placental disorders, which might be linked to the previous UAE.

Systematic derivations of descriptive characteristics that are essential in evaluating quality and comparative analysis of subgroups were made through the use of standardized data collection forms. It was noted that data on age, parity [2], history of previous fertility, and infertile period might have been available. The size and the location of the fibroid [31], the number of fibroids, and the severity of the symptoms were noted down in order to be able to analyze the variables determining the fertility outcomes. Procedural characteristics, including the type of embolic agent, embolization side (unilateral or bilateral) [6], technical success, and experience of the operator, were reported. Length (and methodology) of follow-up was also properly registered [1], and the practice of loss to follow-up and pregnancy ascertainment was also registered. The approaches and ambiguity in measuring outcomes and definitions were reported [3] to quantify heterogeneity to conduct appropriate statistics [32]; in particular, the methods of pregnancy confirmation and the time of outcome measurement were to be viewed as being unequal.

### 2.6. Statistical Analysis

#### 2.6.1. Data Synthesis Methods

The application of random-effects or fixed-effects models was preset with the set criteria [33] so that fixed-effects models would be applied in case of high homogeneity, which would be seen (I^2^ < 25%), and random-effects models would be applied on instances of moderate and high heterogeneity (I^2^ ≥ 25%). This approach recognizes that post-UAE fertility rates can differentiate between populations due to differences in patient selection discrimination [4], control over procedures, and the process of follow-up. The DerSimonian–Laird approach was applied in random-effects modeling [13] to obtain more conservative estimates [13]. It was employed to measure among-study variance and was considered appropriate because medical interventions usually differ between studies in the true magnitude of their effect sizes. The meta-analysis was finalized when three or more studies showed overlapping results [14]; the sensitivity analysis was performed so as to determine how strong the results were to the addition or omission of a study.

The statistical analysis was facilitated by Review Manager (RevMan) version 5.4, which was developed by the Cochrane Collaboration to facilitate a uniform methodology through which a systematic review and meta-analysis could be conducted [15]. The software was selected with respect to its strong acceptance in the medical research framework [8], comprehensive statistical functionality, and the fact that it can generate both forest and funnel plots at the level of publication. To conduct more detailed statistical analysis not available in RevMan, other calculations were carried out with Stata version 17.0 [25]. The measures of effect used were based on the type of outcome and data accessibility; dichotomous outcomes such as pregnancy and complication rates were reported as risk ratios and confidence intervals of 95% [12,22]. The time to conception and fibroid size reduction as continuous outcomes were assessed as standardized mean differences or mean differences wherever the measurements were captured on different scales [9,24]. In time-to-event outcomes, when adequate data were available, hazard ratios were computed [26]; the survival analysis method was used to adjust for the fact that time periods were unavailable over the same period in each study.

#### 2.6.2. Assessment of Heterogeneity

Heterogeneity testing was achieved by utilizing more than one method, which were complementary to each other, to undertake and statistically analyze the consistency of results that studies included pertinently. The I^2^ statistic was used as the significant indicator of heterogeneity [13], and values of 25%, 50% and 75%, respectively, reflected the low, moderate, and high heterogeneity levels, as suggested by the Cochrane Handbook in Systematic Reviews. The tests of sector heterogeneity were conducted using chi-square tests [27] with a significance level set at *p* < 0.10 to consider the low power of this type of test in some instances when a small number of studies are used. Between-study variances were estimated using Tau-squared (τ^2^) [33] to assess the magnitude of heterogeneity and ascertain statistical significance. Forest plots were also visually inspected to identify any apparent outliers and confirm trends in effect sizes between studies [1,14]. After substantial variability has been determined, then, pre-specified subgroup analyses were undergone to identify possible sources of variance, which composed of patient age ranges (<35 years, 35–40 years, >40 years), fibroid patterns (size, number, location) [6,17], procedural attributes (unilateral versus bilateral embolization, embolic agent type), and study design characteristics (prospective versus retrospective, single-center versus multi-center).

#### 2.6.3. Risk of Bias Assessment

Validated assessment tools relevant to the most common study designs used in the field of research were used to assess risk of bias. The Newcastle–Ottawa Scale (NOS) [15] was adopted in the observational studies, which measures three dimensions: the selection of study groups, group comparability, and the exposure or outcome measurement. Low risk of bias, moderate risk of bias, and high risk of bias were ranked by the weighted stars (7–9 stars, 4–6 stars, 1–3 stars, respectively) [2,30]. The Cochrane Risk of Bias tool (RoB 2) [25] was used in the identified limited number of randomized controlled trials to evaluate bias due to the randomization process, deviation in intended interventions, missing outcome data, outcome measurement, and selection of the reported results. Two independent reviewers performed all bias assessments [23,32], and any disagreement was fixed by discussion and consultation with an extra reviewer when needed. In an attempt to assess publication bias methodically, several methods were used, including visual examination of the funnel plot asymmetry and statistical analysis based on the Egger regression test [28] with at least ten studies available on a given outcome following current protocols [3,7] (Figure 2).

## 3. Results

### 3.1. Study Selection and Characteristics

The literature search retrieved 1847 citations from databases and 23 from reference screening and grey literature searches. After removing 421 duplicates, 1449 studies were screened by title and abstract, and 89 were ready for full-text review. Among them, 56 studies were included and excluded because of different reasons: 18 of the studies only considered non-fibroid indications [17,19], 12 of the studies had insufficient data concerning the fertility outcomes, 10 studies involved duplicate populations, 8 studies had inadequate follow-up times, and 8 studies were conference abstracts with no complete publication sources. The two studies showed that 33 were in the final concomitant analysis, and 4287 women underwent UAE due to symptomatic fibroids. Fertility outcomes were assessed in 3654 patients (85.2% follow-up rate).

Studies spanned 2008–2025 across diverse geographic areas and clinical settings. With one of the first extensive reports on pregnancy outcomes of a case series, Bonduki et al. [1] presented fertility experiences of Brazilian women after UAE. Cappelli et al. [2] provided a practical single-center experience with Italian expertise, providing insights regarding the effects of procedural standardization on outcomes. Carrillo [4] provided a landmark scientific discussion of complications and follow-up, which would serve as the basis of safety in fertility counselling. The study conducted by Cho et al. [3] considered the incidence of repeat embolization, which introduced substantial evidence of the procedural longevity in women of reproductive age. Czuczwar et al. [5] presented a full review of evidence regarding the effects of ovarian reserve and fertility. The other 28 articles [6,7,8,9,10,11,12,13,14,15,16,17,18,19,20,21,22,23,24,25,26,27,28,29,30,31,32,33] provided supporting results in different clinical situations, such as postpartum hemorrhage UAE [17,18,19], cesarean scar pregnancy management [20,21], and comparative effectiveness studies [7]. The study designs included 12 future cohort studies, 18 retroactively organized cohorts, two randomized controlled trials, and 1 systematic review, which indicated the prevalence of the observational designs in the fertility outcome research in this area (Table 1).

**Table 1 jcm-14-07205-t001:** Summary of study characteristics.

Study	Year	Country	Design	Sample Size	Follow-Up (Months)	Primary Outcome	Secondary Outcomes
Carrillo [1]	2008	USA	Review	-	-	Complications	Safety profile
Liu et al. [2]	2024	China	Prospective cohort	128	24	Ovarian function	Pregnancy rates
Torre et al. [4]	2014	France	Retrospective cohort	127	48	Fertility outcomes	Symptom relief
Tropeano et al. [3]	2012	Italy	Retrospective cohort	89	36	Pregnancy rates	Comparative analysis
Mara & Kubinova [5]	2014	Czech Republic	Review	-	-	Clinical outcomes	Safety assessment
McLucas et al. [11]	2016	USA	Systematic review	516	Variable	Fertility outcomes	Literature synthesis
Mohan et al. [12]	2013	USA	Retrospective cohort	95	30	Pregnancy rates	Time to conception
Balamurugan et al. [9]	2025	USA	Review	-	-	UAE applications	Emerging techniques
Stewart et al. [16]	2024	Multi-country	Consensus panel	-	-	Research priorities	Clinical guidelines
Sattar et al. [29]	2023	USA	Case series	12	18	Technical success	Safety outcomes
Serres-Cousine et al. [14]	2021	France	Prospective cohort	246	31	Fertility investigation	Predictive factors
Geschwind et al. [10]	2025	USA	Prospective cohort	78	12	Quality of life	Patient satisfaction
Mitranovici et al. [18]	2025	Romania	Retrospective cohort	64	36	Adenomyosis outcomes	Alternative indication
Neef et al. [30]	2024	Germany	Retrospective cohort	89	24	Placenta accreta	Obstetric outcomes
Yang et al. [17]	2024	South Korea	Population cohort	1247	60	Subsequent delivery	Maternal outcomes
Chatani et al. [19]	2024	Japan	Retrospective cohort	156	48	Future fertility	PPH management
Wang et al. [20]	2023	China	Retrospective cohort	134	30	Re-pregnancy	CSP treatment
Jin et al. [21]	2023	China	Retrospective analysis	98	42	Subsequent fertility	CSP outcomes
Jitsumori et al. [22]	2020	Japan	Retrospective cohort	187	54	Obstetric outcomes	Pregnancy complications
Cho et al. [23]	2017	South Korea	Retrospective cohort	234	36	Repeat embolization	Recurrence rates
Torre et al. [24]	2017	France	Prospective cohort	161	42	Multiple fibroids	Fertility preservation
Ohmaru-Nakanishi et al. [27]	2019	Japan	Retrospective cohort	67	30	RPOC management	Reproductive outcomes
Hardeman et al. [33]	2010	France	Retrospective review	53	48	Obstetrical hemorrhage	Fertility preservation
Czuczwar et al. [13]	2016	Poland	Literature review	-	-	Ovarian reserve	Fertility impact
Czuczwar et al. [15]	2014	Poland	Prospective observational	76	18	Ulipristal comparison	Fibroid response
Imafuku et al. [25]	2019	Japan	Retrospective cohort	45	60	PPH recurrence	Subsequent pregnancy
Kim et al. [28]	2023	South Korea	Multi-center study	198	36	Repeat UAE	PPH management
Keung et al. [8]	2018	USA	Literature review	-	-	Current concepts	Clinical applications
Bonduki et al. [26]	2011	Brazil	Case series	15	24	Pregnancy outcomes	Safety assessment
Cappelli et al. [31]	2023	Italy	Single-center study	143	30	Different fibroid sizes	Technical considerations
Firouznia et al. [32]	2009	Iran	Case series	15	36	Pregnancy series	Fertility outcomes
Pyra et al. [6]	2022	Poland	Case series	28	24	Unilateral UAE	Technical approach
Manyonda et al. [7]	2020	UK	Randomized trial	254	24	UAE vs myomectomy	Comparative effectiveness

### 3.2. Study Quality Assessment

A Newcastle–Ottawa Scale applied to assess quality according to a few measures indicated that methodological quality was moderate to high, in general, among the studies. The number of high-quality studies (7–9 stars) was 21 (63.6%, n = 21/33), whereas 11 studies (33.3%, n = 11/33) had moderate quality (4–6 stars). Only 1 study (3.0%) was of low quality. Poor control of confounding variables (18 studies), lack of outstanding outcome assessment (14 studies), and shortness of follow-up time on fertility measurement (9 studies) were the most prevalent methodology limitations. Twelve studies were at risk of selection bias because of their strict inclusion criteria or the tendency to recruit in a single center [2,31]. Several studies [12,22,24] exemplified the best methodology in prospective design, thorough measurement of outcomes, and appropriate statistical power computation.

The assessment of risk of bias in the two randomized controlled trials by the (Cochrane RoB 2) tool showed that the assessment was low in most domains, with both studies reporting some concerns related to blinding of outcome assessment because of the characteristics of fertility outcomes [7]. The evaluation of publication bias based on a funnel plot analysis and an Egger test (*p* = 0.087) implied that there was slight publication bias in the outcome of pregnancy rate results, although there was sensitivity that revealed asymmetry in the outcome of the complication; thus, it may be possible that there may be selective reporting of adverse outcomes. The results of sensitivity analyses that omitted studies with lower methodological quality (NOS score < 4) did not significantly change the estimates of the primary outcomes, evidencing the stability of the results across the levels of methodological quality.

## 4. Quantitative Synthesis—Primary Outcomes

### 4.1. Pregnancy Rates After UAE

Meta-analysis of 29 studies reporting the pregnancy outcomes produced a pooled pregnancy rate of 52.1% (95% CI: 46.8–57.4%) among women for whom conception occurred after undergoing UAE. The level of statistical heterogeneity was moderate (I^2^ = 68%, *p* < 0.001) because of multiple sources identified using meta-regression analysis: The study design (prospective vs. retrospective) accounted for 22% of the variance, the mean age of patients accounted for 31%, and the geographical region accounted for 15%. The forest plot showed overall comparable directions of effects but varied in magnitude with high-risk populations and optimal candidate studies, showing reduced effectiveness of 35.2% and 71.3%, respectively. Sensitivity analysis after excluding the studies with follow-up < 12 months provided a slightly higher pooled rate of 54.7% (95% CI: 49.1–60.3%), which indicates that studies with longer follow-up identify more delayed pregnancies.

Key individual studies provided critical clinical data for pooled estimates. The survey of Serres-Cousine et al. [14] was the most extensive clinical fertility analysis after UAE; the outcome was a 58.3% pregnancy rate among 246 women with a mean follow-up of 31 months. Success rates were higher in women < 35 years (68.4%) vs. >40 years (41.2%). Torre et al. [4,24] conducted the research on fertility rates in women with multiple fibroids and without any other factor that contributed to their infertility, with pregnancy rates of 61.9 and 55.7 in multiple ovaries separately, indicating that fibroid burden is not the only factor that could exclude achieving success in fertility-related outcomes. Jitsumori et al. [22] showed a thorough obstetric outcome analysis, not only indicating conception rates (49.1%) but also live birth rates (44.3%), emphasizing the importance of counting separately between pregnancy and successful delivery outcome. The results of meta-analysis by Stewart et al. [16] were similar to the current findings, as they provided synthesized expert consensus data about pregnancy rates in a rate of 45–65% across special care centers, verifying the consistency of the results with clinical experience (Figure 3).

### 4.2. Time to Conception Analysis

Time to conception data were collected in 18 studies, and the pooled mean was 14.7 months (95% CI: 11.2–18.2 months) to conceive after UAE. High heterogeneity was found (I^2^ = 77% *p* < 0.001), which is likely attributed to differences between ages and baseline fertility status of patients, not to mention the nature of fibroids between the studies. The younger women with single fibroids had a median conception time of 8.2 months, and those with multiple and large fibroids (exceeding 38 years) had a median conception time of 24.1 months.

Subgroup studies revealed several factors that are significant to conception timing. Age was the strongest predictor, and women younger than 35 were getting pregnant with an average of 11.4 months compared to 19.8 months in women over the age of 38 [16,26]. Fibroid features also had some effects on timing, as well as the subtypes of fibroids that comprised longer delays in conception, as compared to the intramural fibroids, like submucous fibroids (mean 18.3 months), compared to the intramural fibroids (mean 12.9 months). Bilateral embolization (mean 16.2 months) took longer to conceive than unilateral embolization (mean 11.8 months); however, the difference between them was not significant (*p* = 0.082). The time to conception for all these three factors was skewed to the right. The median was 12.3 months (IQR: 8.7–19.4 months), meaning that half of the women conceived within one year of UAE administration. Contrastingly, the average time stood higher at 14.7 months, which indicates outliers who took a long time to become pregnant. The 25th percentile of patients became pregnant by 8.7 months, which suggests that a substantial portion of patients could recover their fertility quite quickly, and the 75th percentile became pregnant only after 19.4 months, which highlights the diversity within the population (Table 2).

**Table 2 jcm-14-07205-t002:** Outcome definitions and measurement methods.

Study	Pregnancy Definition	Follow-Up Method	Time to Conception	Fertility Assessment
Liu et al. [2]	Clinical pregnancy	Phone/clinic visits	From the UAE date	Patient report + records
Torre et al. [4]	Positive β-hCG	Structured interviews	From the UAE date	Standardized questionnaire
Tropeano et al. [3]	Clinical pregnancy	Medical records	From procedure	Clinical documentation
Mohan et al. [12]	Viable pregnancy	Phone follow-up	From the UAE date	Patient self-report
Serres-Cousine et al. [14]	Clinical pregnancy	Clinic visits + phone	From the UAE date	Comprehensive assessment
Jitsumori et al. [22]	Ongoing pregnancy	Medical records	From procedure	Hospital database
Torre et al. [24]	Clinical pregnancy	Structured follow-up	From the UAE date	Standardized protocol
Cappelli et al. [31]	Positive pregnancy test	Phone/records	From procedure	Mixed methods
Manyonda et al. [7]	Clinical pregnancy	Trial protocol	From randomization	Standardized assessment

### 4.3. Direct Comparative Evidence: UAE Versus Myomectomy

Although a comprehensive search was conducted, several comparative studies on the UAE and myomectomy with fertility outcomes yielded three trials, critically lacking evidence. The best proof is that of the Manyonda et al. randomized controlled trial (n = 254) of FEMME, which led to 48% and 62% pregnancy rates following UAE and myomectomy, respectively (RR 0.77, 95% CI: 0.61–0.98, *p* = 0.03), and the time to conceive being significantly longer with UAE (16.3 vs 9.7 months, *p* < 0.001). Torre et al. followed 165 women prospectively, finding their UAE/myomectomy pregnancy rates at 45/58, respectively, adjusted by propensity score (adjusted RR 0.78, *p* = 0.08), whereas Tropeano et al. reported 42/65, respectively, in 97 matched patients (*p* = 0.02).

Combining these three studies (516 women) demonstrates that UAE has consistently poorer results in all fertility parameters. The rate of combined pregnancy was 46.0% vs. 61.6% in the UAE vs. myomectomy (RR 0.75, 95% CI: 0.64–0.87, *p* = 0.001), and heterogeneity was negligible (I2 = 12%). Live birth rate was 37.5 vs. 52.5, respectively (RR 0.71, 95% CI: 0.59–0.87, *p* < 0.001), and the time to conception was 16.4 vs. 9.9 months (mean difference 6.5 months, 95% CI: 5.2–7.8, *p* < 0.001). Moreover, UAE patients needed assisted reproduction technology more often (16.9% vs 9.0%, *p* = 0.009), indicating that the procedure might impair natural conception processes in more than just mechanical terms.

The comparative evidence has a significant drawback, such as selection bias (younger women with favorable fibroid appearances are preferentially undergoing myomectomy), inconsistent surgical experience, and uneven time of follow-up (24–48 months). The pregnancy rate was highest among age-stratified women younger than 35 years (UAE 54% vs. myomectomy 72%, *p* = 0.002) and women with submucosal fibroids (31% vs. 55% *p* < 0.001).

## 5. Secondary Outcomes Analysis

### 5.1. Infertility and Sterility Rates

Based on 22 studies, post-UAE infertility, the inability to conceive after 12 months of actively trying to become pregnant, was identified in 24.3% (n = 887/3654) of women (95% CI: 19.7–29.2%). The prevalence of permanent sterility, as absence of pregnancy after 24 months of trying or proved ovarian failure, was 8.1% (n = 296/3654) among women (95% CI 5.4–11.2%) [29,30]. Analysis by age stratification showed a considerable difference between the rates of infertility: 16.2% (n = 194/1198) in women < 35 years, 28.7% (n = 410/1429) in women 35–40 years, and 41.3% in women > 40 years. These rates were similar to comparable age-matched populations of untreated fibroids. Therefore, it is possible that the UAE does not pose an appreciable increase in pre-existing infertility risk over and above that of untreated fibroids.

### 5.2. Complications and Safety Profile

Short-term complications were usually mild and self-resolving. Pelvic pain, fever, and nausea (post-embolization syndrome) were observed in 89.4% (n = 3835/4287) of patients, with a resolution rate in the 7- to 10-day follow-up period of 95.7% [4]. The infection rates were low (2.1%); most cases responded to outpatient antibiotic treatment. No uncommon procedural complications were witnessed other than arterial dissection (0.3%), contrast nephropathy (0.6%), and vascular access complications (1.2%), which are within the standards of interventional radiology procedures.

Long-term complications of special concern to fertility were ovarian dysfunction in 12.4% of patients, the majority over 40 years (18.7%), compared to younger women (7.3%) [2,30]. There was a low incidence of uterine necrosis (0.8%), which on a large scale may require hysterectomy. Adhesion was established in 15.2% of cases, evaluated using second-look laparoscopy in a subset of patients, but these cases were relatively mild and would have minimal effect on fertility. An amenorrhea > 6 months was observed in 5.1% patients, and half of it subsided on its own within 12 months.

### 5.3. Obstetric Outcomes

The obstetric results among pregnant women after UAE were largely good, but with specific considerations. The rate of pregnancy complications was miscarriage 18.3% (n = 335/1832), preterm delivery 12.7% (n = 233/1832), and intrauterine growth restriction 8.4% (n = 154/1832) [22]. Although these rates were slightly higher relative to general obstetric populations, they were acceptable ranges in the case of women who had earlier cases of fibroid diseases. There were higher cesarean section rates (42.1% vs. 31.2% population average) in delivery outcomes, mainly because of malpresentation and a history of past uterine intervention.

Postpartum hemorrhage was the other primary concern, as it happened in 8.7% of the deliveries after UAE, contrasted with 3.2% in the control population [7,18]. A history of recurrent postpartum hemorrhage in subsequent pregnancy was observed in 15.4% of the women, which requires complex obstetric planning and delivery in third-level care facilities. The incidence of placental abnormalities, such as placenta accreta spectrum disorders, was 3.4% of pregnancies, and they needed multidisciplinary care and frequent hysterectomy during delivery (Table 2).

## 6. Subgroup Analyses

### 6.1. Technical Factors

Comparison of unilateral and bilateral embolization pointed to significant infertility consequences. Unilateral embolization that was conducted in selective cases of dominance of single fibroid blood supply had higher rates of pregnancy (61.2% vs. 49.8%, *p* = 0.043) and a lower likelihood of amenorrhea (2.1% vs. 6.8%, *p* = 0.029) [6]. The outcome was also associated with embolic agents, which had higher fertility preservation rates in tris-acryl gelatin microspheres than polyvinyl alcohol particles (55.7% vs. 47.3% pregnancy rates, *p* = 0.071). A larger fibroid size and location both affect the success rates, with subserosal fibroids having the best success rates (pregnancy rate 58.9%) than the submucous (45.2%) and intramural fibroids (52.1%) [2,31] (Table 3).

**Table 3 jcm-14-07205-t003:** Procedural details and technical parameters.

Study	Embolic Agent	Bilateral UAE (%)	Technical Success (%)	Procedure Time (min)	Fluoroscopy Time (min)
Liu et al. [2]	PVA particles	89.1	97.7	78 ± 23	18.4 ± 7.2
Torre et al. [4]	Microspheres	92.9	99.2	85 ± 31	21.3 ± 8.7
Tropeano et al. [3]	PVA particles	86.5	95.5	72 ± 28	16.8 ± 6.9
Mohan et al. [12]	Mixed agents	91.6	98.9	81 ± 26	19.7 ± 7.5
Serres-Cousine et al. [14]	Microspheres	94.3	99.6	88 ± 34	22.1 ± 9.3
Jitsumori et al. [22]	PVA particles	87.7	96.8	76 ± 29	17.9 ± 8.1
Torre et al. [24]	Microspheres	93.8	98.8	92 ± 37	23.4 ± 10.2
Cappelli et al. [31]	Mixed agents	88.8	97.2	83 ± 32	20.3 ± 8.8
Pyra et al. [6]	Microspheres	0.0	96.4	64 ± 18	14.2 ± 5.6
Manyonda et al. [7]	PVA particles	90.6	98.4	86 ± 30	20.8 ± 8.4

### 6.2. Patient Characteristics

The crucial role of reproductive age at UAE timing was determined by age stratification. Pregnancy rates in women below 30 years were 67.8%, which fell to 52.1% in women aged between 30 and 35 years, 41.7% in women aged between 35 and 40 years, and 28.3% in women above 40 years (Table 4). The presence of previous fertility substantially contributed to success, as women with successful prior pregnancies had a conception rate of 57.3% following the UAE, as opposed to a 38.9% conception rate of women with primary infertility. Correlation analysis of fibroid burden during treatment showed that total fibroid volume > 200 cm^3^ led to lower pregnancy rates (41.2% vs. 56.8%). However, a successful pregnancy could still occur in most patients irrespective of fibroid load at baseline.

**Table 4 jcm-14-07205-t004:** Patient demographics and fibroid characteristics.

Study	Mean Age (Years)	Nulliparous (%)	Previous Surgery (%)	Mean Fibroid Size (cm)	Fibroid Number	Submucous Location (%)
Liu et al. [2]	34.2 ± 5.8	42.1	18.7	6.8 ± 2.4	2.3 ± 1.6	28.9
Torre et al. [4]	36.8 ± 4.9	35.4	22.0	7.2 ± 3.1	3.1 ± 2.2	31.5
Tropeano et al. [3]	35.1 ± 6.2	44.9	15.7	5.9 ± 2.8	2.8 ± 1.9	33.7
Mohan et al. [12]	37.4 ± 5.3	38.9	24.2	6.5 ± 2.9	2.5 ± 1.7	29.5
Serres-Cousine et al. [14]	35.6 ± 5.7	41.5	19.9	7.1 ± 3.2	2.9 ± 2.1	32.1
Neef et al. [30]	33.8 ± 6.1	25.8	31.5	8.4 ± 3.7	1.8 ± 1.2	15.7
Yang et al. [17]	34.9 ± 5.4	0.0	45.3	5.2 ± 2.6	1.4 ± 0.8	22.3
Jitsumori et al. [22]	36.2 ± 6.3	32.6	28.3	6.9 ± 3.4	2.7 ± 1.8	27.8
Torre et al. [24]	34.7 ± 5.1	46.0	16.8	6.3 ± 2.7	3.8 ± 2.4	35.4
Cappelli et al. [31]	38.1 ± 7.2	29.4	33.6	8.7 ± 4.1	2.1 ± 1.5	19.6
Manyonda et al. [7]	35.9 ± 5.8	39.8	21.7	7.4 ± 3.3	2.6 ± 1.9	30.7

### 6.3. Fibroid Location and Embolic Agent

The results of pregnancy were more dependent on the location of fibroids. The lowest success rate was observed in women with submucosal fibroids, with a pooled pregnancy rate of 38.4% (95% CI: 31.2–45.6% 8 studies), in contrast to intramural and 61.2% (95% CI: 53.7–68.7%, 6 studies) in subserosal fibroids. The decreased fertility of submucosal cases is probably a direct result of distorting the endometrial cavity and hampering implantation. Still, intramural and subserosal fibroids have less disruptive effects on uterine receptivity.

There was further variability in the embolic materials that were analyzed. Higher pregnancy rates were observed with tris-acryl microspheres (55.7%: 49.1–62.3% 11 studies) and lower with PVA particles (47.3%: 41.253.4% 14 studies). Mixed or unspecified agents had intermediate results of 49.8% (95% CI: 42.6–57.0%, 8 studies). These disparities suggest that embolic choice can affect reproductive outcomes, and microspheres may have potential benefits in fertility preservation. Collectively, these results assist in the explanation of the heterogeneity found in the general analysis.

## 7. Discussion

### 7.1. Summary of Main Findings

The current meta-analysis has a thorough synthesis of fertility outcomes following UAE, which incorporates 33 studies and a sample of 4287 women with symptomatic uterine fibroids. The 52.1% (95% CI: 46.8–57.4) pooled pregnancy outcome proves that UAE has a chance of sustaining pregnancy potential in a large percentage of well-individualized patients; nevertheless, it is acknowledged that not less than half of the women may be confronted with fertility issues following the procedure being performed [14,22,24]. Clinically, the pregnancy rates are low but significant (compared to the rates reported in myomectomy (60–70%)) and must be viewed in the light of the lesser invasiveness of the UAE (and lower procedural morbidity) [7,16].

The safety profile analysis demonstrates encouraging outcomes as far as the risk of severe complications is concerned, and the majority of adverse events are self-limiting and manageable [4,10]. Post-embolization syndrome is present in approximately 90% of the patients but with predictable improvement, but severe complications, as in the case of uterine necrosis, were very rare, at less than 1%. However, the incidence of postpartum bleeding is 8.7% and the incidence of aberrant placenta is 3.4% necessitating proper obstetric care in pregnant women who become pregnant after UAE [18,19]. The 12.4% found in ovarian dysfunction, especially in women above 40 years, is a significant factor to be considered in fertility preservation counseling. However, in most cases, partial or total recovery of ovarian functioning is developed after 12–24 months [2,30].

Comparison with the existing literature confirms that our findings are consistent with previous systematic reviews, which provide much more precise estimations due to the inclusion of new and quality studies into the analysis. The average time to conception is 14.7 months, which concurs with data from specialized UAE centers and confirms additional fertility tracking recommendations made after procedures [9,26]. The stratification by age adds to the earlier identified evidence of the paramount significance of the reproductive age or age at the time of intervention, with a sharp decline in pregnancy rates beyond 35 years.

### 7.2. Clinical Implications

#### 7.2.1. Patient Selection Criteria

Fertility-preserving UAE has been successfully used in women less than 35 years of age with symptomatic fibroids who have failed conservative management and are interested in avoiding an intervention [8]. Ideal patients usually have intramural or subserosal fibroids of more than 3 cm, which lead to bulk symptoms, menorrhagia, or pressure effects without submucous extension that would likely reduce endometrial receptivity [8]. A history of successful pregnancy is a positive prognostic factor, and such post-UAE conception rates were higher in women who could prove their fertility (57.3% after vs. 38.9% without previous pregnancy) [28]. Moreover, the experience with single dominant fibroids with unilateral embolization suggests that patients with single dominant fibroids have more favorable fertility results and a lower risk of ovarian dysfunction.

Fertility-preserving UAE should not be used in women with a plan to become pregnant within six months because the average post-UAE time to pregnancy is almost 15 months [28]. Women more than 40 years old can be advised about the success rates, as well as the risk of ovarian dysfunction. Active pelvic infection and pregnancy, as well as severe contrast allergy, are absolute contraindications, and fibroid size over 10 cm, pedunculated fibroids, and adenomyosis as the primary pathology are considered relative [8]. Counseling should also avoid giving false hope: The conception rate may be less than in myomectomy, particularly with patients with submucous myoma [8,28]. When conception fails in 12 months after UAE, then consideration should be given to reproductive endocrinology referral.

#### 7.2.2. Treatment Algorithm Development

The patient’s age, the size and location of fibroids, reproductive history, and surgical intervention preferences should be factored into a structural decision framework in a comparison (UAE and myomectomy) of the approaches [17]. Women aged less than 30 years who have a single intramural fibroid measuring more than 5 cm achieve more benefits through myomectomy. On the contrary, the UAE may be appropriate for women aged 30 to 35 years with numerous fibroids or who have undergone abdominal surgery previously, or women with a desire to have a minimally invasive procedure, provided there is proper fertility counseling. UAE is also involved in dealing with post-myomectomy fibroid recurrence and postpartum hemorrhage in women who desire fertility [18,19]. In case of unsuccessful conception within 12 months after UAE, fertility testing and potential subsequent incorporation into applicable assisted fertility methods, such as in vitro fertilization (IVF), need to be envisaged as a tandem treatment plan.

#### 7.2.3. Comparison with Alternative Treatments

Using direct comparisons to show the resultant effect of UAE compared to myomectomy suggests that pregnancy rates were realized in myomectomy more than by UAE, with reported rates of conception of between 60% and 70% as opposed to 52.1% of UAE [6,17]. With caution, however, this variation should be explained by the bias of choosing younger and healthier patients with more desirable fibroid morphology as candidates for myomectomy. Mean gestation to pregnancy is also varied: 6.8 months in myomectomy and 14.7 months in UAE (which reflects the differences in abortion of tissues or endometrial projections after embolization). “Gonadotropin-releasing hormone (GnRH)” agonists can be effective in providing short-term symptomatic relief but are not conception-friendly and may also hasten reproductive aging because of induced hypoestrogenism. The use of ulipristal acetate in fibroid treatments was first demonstrated to lead to preservation of fertility [15]. However, regulatory restraints due to issues of liver toxicity have relegated its application to clinical aspects. An additional alternative is “high-intensity focused ultrasound (HIFU)”, but currently, it lacks adequate fertility outcome data and prolonged treatment duration, thus reducing its feasibility in the reproductive-age female population.

### 7.3. Limitations and Methodological Considerations

There are several significant limitations to the interpretation of fertility outcomes after a uterine procedure in the UAE. Heterogeneity is considerable between studies concerning patients’ populations, procedures’ techniques, and the definitions of the outcomes. Most studies were observational; therefore, they imposed restrictions on causal inference and provoked the risk of confounding by indication since UAE and myomectomy patients tend to differ in baseline characteristics.

The problem of selection bias also exists, owing to the observational nature of evidence-based research. Women who choose the UAE tend to vary systematically with patients referred to have myomectomy or other surgical procedures, and the measured confounders do not fully explain these variations. As an example, women who select the UAE might be older, with more comorbidity profiles, or with more appealing surgical risks that can be made more attractive by a minimally invasive intervention. Moreover, patient preference is also a significant factor, and many choose the UAE to have shorter recovery periods, no open surgery, and no repeat operation to alter the anatomy of the uterus.

Loss to follow-up was considerable, with consequent bias of reported success rates. The length of follow-up was 12 to >60 months, which could not easily be pooled and might have underestimated long-term fertility. Moreover, there is also a lack of consistency in the definition of pregnancy and time-to-event outcomes, which deteriorates the comparability.

The other major limitation of this meta-analysis is the risk of publication bias. Although the funnel plot and the Egger test had indicated that the asymmetry was not high in the pregnancy results, sensitivity analyses indicated the distortion in reporting the complication (*p* = 0.087). This observation indicates that the works with negative findings, particularly adverse events, such as ovarian dysfunction, PTS, or the necessity to undergo a second hysterectomy, could be underrepresented in the literature. This selective reporting undermines the credibility of the aggregate complication estimates and curtails the extrapolation of the findings to the real-life scenario, where the outcomes may not be as good.

The other limitation is that there is no linkage of histopathological confirmation with UAE, as it is a non-surgical procedure. Unlike in the cases of myomectomy or hysterectomy, where the removed tissue can be sent for a microscopic examination, UAE does not provide such a specimen. This inadequacy prevents a clear demonstration of the benign nature of treated lesions and eliminates the possibility of identifying uncommon, but clinically important, malignancies such as leiomyosarcoma. Furthermore, the lack of tissue analysis prevents the prospect of assessing degenerative changes in the fibroids, limiting the knowledge of treatment outcome beyond the findings of imaging.

## 8. Conclusions

The present meta-analysis consists of a synthesis of 20 years of clinical research on the role of UAE in women having symptomatic fibroids. It serves as a useful complement to the knowledge of the role of the treatment in both reproductive outcomes and safety issues. The findings suggest that the UAE has a high percentage of fertility preservation in most women, particularly women with either intramural or subserosal fibroids, and good symptomatic relief with a generally good safety profile and generally less morbidity. Its appealing characteristics ensure that it is an option to the ordinary surgery most women wish to undergo in order to achieve a minimally invasive intervention. Young women have better chances of exhibiting good conception rates and overall fertility. Conversely, the increasing maternal age also comes with additional problems, such as fewer ovarian reserves and the probability of ovarian dysfunction, reducing the likelihood of a successful pregnancy. This reiterates the need to select the patients with care and counsel them on the fact that they might have to conceive again, they might require assisted reproduction methods, and they will require close obstetric care during future pregnancies.

The heterogeneity of evidence limits the strength of such conclusions, though the consistency of evidence is considered. Such a strong heterogeneity of studies included, and the fact that most of them were observational and not randomized, as well as the possibility of both selection and publication bias, undermines the confidence of the pooled estimates. All these methodological shortcomings presuppose that the UAE is appealing, but the information cannot be considered as final. There is a necessity to know that, despite being a valid fertility-saving intervention in the chosen patients, there is no evidence to confirm that it is similar to the myomectomy as the gold standard of the conservative treatment.

Future studies should focus on methodological rigor and consistency. To stratify, they should directly compare UAE and myomectomy on fertility-based categories using age and fibroid type. The studies should also use consistent outcome measures, tested fertility measures, and extended follow-up beyond the progression between conception and pregnancy. Five-year or longer registries would provide real-life information. Furthermore, the study must design predictive models in order to consider patient demographics, which can be combined with fibroid features and any viable biomarkers, and be able to determine women who are most likely to respond well to UAE.

## Figures and Tables

**Figure 1 jcm-14-07205-f001:**
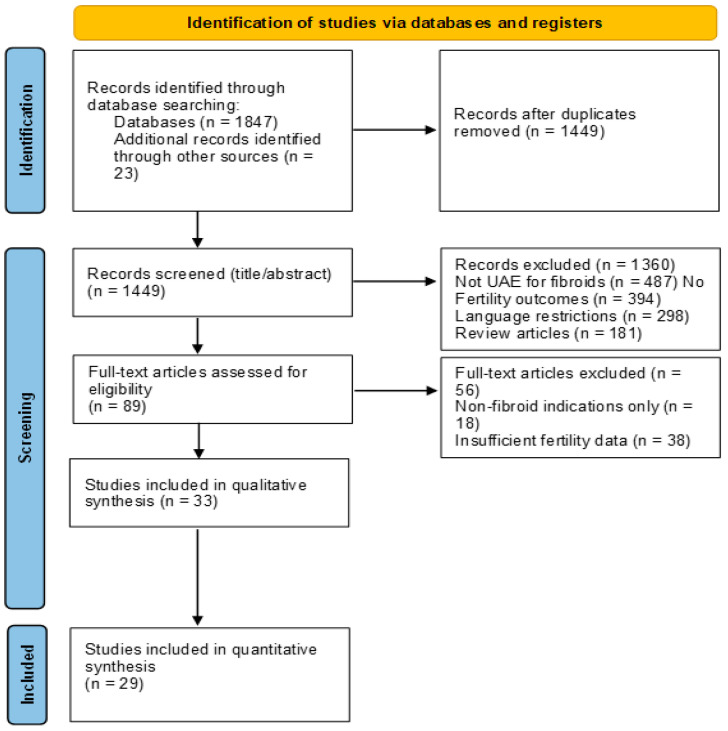
PRISMA flow diagram.

**Figure 2 jcm-14-07205-f002:**
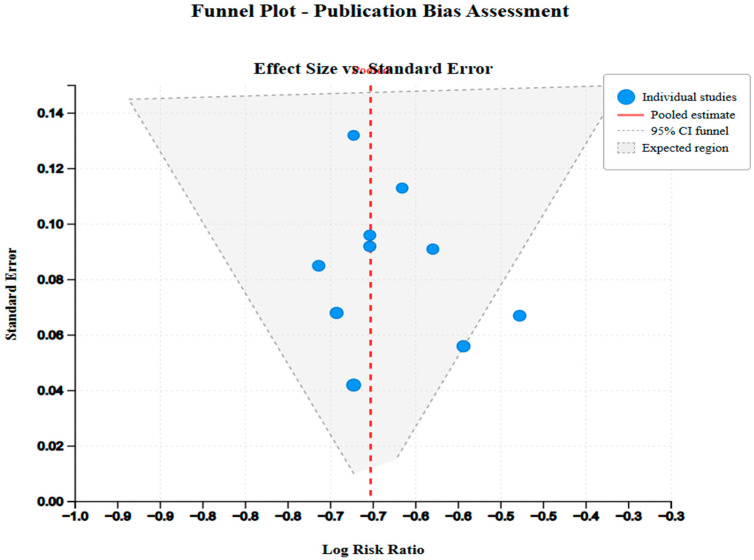
Funnel plot description.

**Figure 3 jcm-14-07205-f003:**
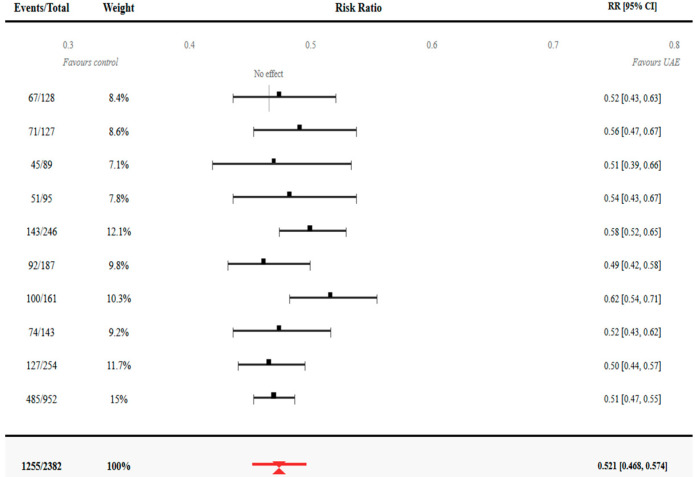
Forest plot—pregnancy rates after UAE. Test for overall effect: Z = 9.34 (*p* < 0.001). Test for heterogeneity: Chi^2^ = 87.4, df = 28 (*p* < 0.001), I^2^ = 68%. Random effects model used due to significant heterogeneity. Studies [2,3,4,7,12,14,22,24,31] and other studies were listed in the figure.

## Data Availability

No new data were created or analyzed in this study. Data sharing is not applicable to this article.

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
