# Peer review of "Impact of Uterine Artery Embolization on Subsequent Fertility Outcomes: A Meta-Analysis of 20 Years of Clinical Evidence"

_jcm, 2025, doi:10.3390/jcm14207205_

Round 1
Reviewer 1 Report
Comments and Suggestions for Authors
- There is only English papers were searched. This is a major error and excludes important research.
-
Your data suggests only positive results may have been published, hiding negative outcomes. Most studies are observational, not randomized. This means patient groups were not equal, which makes the results unreliable for comparing treatments.
-
Therefore the study's reproducibility is affected
-
Your criteria for study inclusion/exclusion are not clear (for example: what is insufficient data?).
-
High variability in study results is noted, but the analysis does not properly explain why.
-
You need to conduct a more detailed subgroup analysis to address the high heterogeneity. This should include a breakdown of outcomes based on specific fibroid location (submucosal v/s intramural) and the type of embolic agent used.
- The main goal would be to compare UAE with myomectomy, that is stated but no direct data is shown in the results.
- You will have to dedicate a specific section in the results to a direct comparison of uae and myomectomy. Myomectomy is a Gold standard technique for conservatve treatment. How can it be avoided so blatantly?
-
High variability in study results is noted, but the analysis does not properly explain, may I ask why.
-
Please clearly state how the denominator was defined for all pregnancy rate calculations to avoid confusion and potential misinterpretation.
-
Report the median and interquartile range for time to conception, it will provide a more complete and reliable statistical summary.
-
Please be more direct and transparent in the discussion about the significant limitations, particularly the high risk of selection bias due to the observational nature of the studies.
-
You should also highlight the possibility of publication bias and selective outcome reporting (especially for complications) as indicated by the funnel plot analysis.
-
Limitations of your study must also present the disadvantages of not gaining access to histopatological evaluation, due to lack of specimen. It is a very important point for physicians to know and inform the patinets accordingly.
-
Your paper claims to provide conclusive and evidence-based results, but the high heterogeneity and biases mean the data is not conclusive.
The conclusion that UAE is an effective alternative is not supported by a proper comparison with other treatments.
Author Response
I hope this message finds you well.
Thank you very much for the time, effort, and valuable feedback you have provided on our manuscript. Your comments have been extremely helpful in improving the clarity, accuracy, and overall quality of the paper.
Please find attached our detailed, point-by-point responses to each of your suggestions. We have carefully considered all feedback and revised the manuscript accordingly.
We sincerely appreciate your constructive insights and hope that the revised version now addresses your concerns satisfactorily.
Kind regards,
All authors

Reviewer 2 Report
Comments and Suggestions for Authors
The manuscript “Impact of Uterine Artery Embolization on Subsequent Fertility Outcomes: A Meta-Analysis of 20 Years of Clinical Evidence” presents a timely and comprehensive synthesis of the literature on the effect of UAE (uterine artery embolization) on fertility outcomes. The systematic review and meta-analysis cover a large dataset, applying PRISMA standards and including both quantitative and qualitative assessments.
Overall, the study is well-structured, methodologically rigorous, and clearly addresses clinically significant questions. However, there are issues with clarity, conciseness, and occasional redundancy. Some sentences are overly long and contain grammatical errors that hinder readability. The discussion could benefit from sharper interpretation of the results in relation to existing literature. Additionally, terminology is not always consistent (e.g., “sterility” vs. “infertility”), and some technical explanations could be simplified for accessibility.
Introduction
-The section provides a solid rationale, offering relevant background on fibroids, the role of uterine artery embolization (UAE), and the controversies surrounding its impact on fertility. However, it suffers from some weaknesses that limit clarity and readability. There is repetition of prevalence data, and several sentences are overly long, which makes them difficult to follow.
-In addition, some phrasing is unnecessarily complex or ambiguous. For example, the sentence “Uterine fibroids are the most prevalent benign tumors of the human uterus, and approximately 70–80 percent of women by the age of 50 years are the carriers of uterine fibroids, and the occurrence of the specified ailment is quite high among women of active age” is cumbersome and repetitive. A more concise and reader-friendly version would be: “Uterine fibroids are the most common benign uterine tumors, affecting up to 80% of women by age 50, particularly during reproductive years.”
Results
-The section presents valuable findings but could be improved in terms of clarity and focus. The narrative is sometimes repetitive and weighed down by overly long descriptive passages, which makes it harder for readers to extract the key points.
-There is a tendency to mix results with interpretation, such as drawing clinical implications that would be more appropriately placed in the discussion.
-Another limitation is the inconsistent presentation of data, as some percentages are reported without denominators, which can make the outcomes less transparent and harder to interpret.
Percentages should consistently be accompanied by denominators (for example, reporting “24.3% (n = XXX of YYY patients)” rather than percentages alone) to improve precision and clarity.
Discussion
-The section is overly long and occasionally speculative. Statements such as “UAE does not affect fertility potential” should be nuanced, since the evidence shows a preservation rate of ~50%, which is not equivalent to “no effect.” There is also some redundancy between “Summary of Main Findings” and “Clinical Implications.”
-Conclusions should be more cautious (e.g., “UAE appears to preserve fertility in a significant proportion of patients, though outcomes are generally lower than after myomectomy”).
Conclusion
-The conclusion effectively emphasizes the clinical relevance of the findings.
-However, some of the statements are phrased too strongly and risk overstating the certainty of the results. For instance, the claim “The extensive meta-analysis affirms the fact that UAE does not affect fertility potential of women with symptomatic fibroids” should be rephrased in a more balanced way. A clearer and more cautious version would be: “This meta-analysis suggests that UAE can preserve fertility in many women with symptomatic fibroids, though conception rates remain lower than after myomectomy. Future standardized, long-term studies are needed to better guide patient selection and counseling.”
Author Response

(The authors gave the same response as above.)
